# Synthesis of Dihydropyrano[3,2-*c*]pyrazoles via Double Bond Migration and Ring-Closing Metathesis

**DOI:** 10.3390/molecules24020296

**Published:** 2019-01-15

**Authors:** Yoshihide Usami, Kodai Sumimoto, Azusa Kishima, Yuya Tatsui, Hiroki Yoneyama, Shinya Harusawa

**Affiliations:** Department of Pharmaceutical Organic Chemistry, Osaka University of Pharmaceutical Sciences, 4-20-1 Nasahara, Takatsuki, Osaka 569-1094, Japan; e12241@gap.oups.ac.jp (K.S.); e13307@gap.oups.ac.jp (A.K.); e18902@gap.oups.ac.jp (Y.T.); yoneyama@gly.oups.ac.jp (H.Y.); harusawa@gly.oups.ac.jp (S.H.)

**Keywords:** dihydropyrano[3,2-*c*]pyrazole, synthesis, double bond migration, ruthenium hydride catalyst, ring-closing metathesis

## Abstract

Three types of pyrazole-fused heterobicycles, i.e., 1,5-, 1,7-, and 2,5-dihydropyrano[3,2-*c*]pyrazoles, were synthesized from 4-allyloxy-1*H*-pyrazoles. A sequence of the Claisen rearrangement of 4-allyloxy-1*H*-pyrazoles, ruthenium-hydride-catalyzed double bond migration, *O*-allylation, and ring-closing metathesis was employed in this study.

## 1. Introduction

The synthesis of substituted or functionalized pyrazoles has been studied extensively thus far because they show or are expected to show important and diverse bioactivities [1,2]. Celecoxib, a non-steroidal anti-inflammatory drug (NSAID), is a representative pyrazole-containing compound, which acts through selective cyclooxygenase (COX)-2 inhibition. Whereas the late-stage construction of a pyrazole ring through some cycloadditions of already-substituted components is the basis for most syntheses of substituted pyrazoles [3,4], direct functionalization of pyrazoles has not been investigated satisfactorily to date. As investigations on it seem rare, we have been interested in and studied the direct functionalization of pyrazoles through coupling reactions of halogenated analogues derived from commercially available pyrazole [5,6,7,8]. In addition, pyrazole-fused heterocycles have recently been synthesized for reasons similar to those described above or because of characteristic activities not seen in monocyclic substituted pyrazoles [9]. Many pyrazole-fused heterocyclic compounds possess unique and important biological activities [10]. Some examples of pyrano[2,3-*c*]pyrazoles [11,12,13,14], pyrano[3,2-*c*]pyrazoles [15,16], and furo[3,2-*c*]pyrazoles [17,18] are presented in Figure 1. 

The Claisen rearrangement followed by ring-closing metathesis (RCM) is an effective sequence for constructing various polycyclic systems [19,20]. On the basis of our previous work on the synthesis of withasomnines [21,22], we recently reported the synthesis of dihydrooxepino[3,2-*c*]pyrazoles (**4** and its isomers) via a combination of the Claisen rearrangement of 4-allyloxy-1*H*-pyrazoles (**1a**–**d**), *O*-allylation of Claisen rearrangement product **2** into **3**, and subsequent RCM of **3** [23]. This realized the construction of pyrazole-containing 5,7-bicyclic system **4**, shown in Scheme 1.

After the migration of the double bond in the side chain of intermediate **2** in Scheme 1, expected product **5** can be *O*-allylated to **6**. The subsequent RCM of **6** may provide a pyrazole-containing 5,6-bicyclic system, i.e., a dihydropyrano[3,2-*c*]pyrazole. These are expected to show various types of activities. There have been many reports of syntheses of pyrano[2,3-*c*]pyrazoles [10,11,12,13,14], but very few for pyrano[3,2-*c*]pyrazoles [15,16,24,25]. In addition, the development of a new synthetic method for furo[3,2-*c*]pyrazoles, which are extremely important as mentioned above, seems possible if both double bond migration and dehydrohalogenation occur on a 5-allyl-4-(2-haloethyl)oxy-1*H*-1-tritylpyrazole. Described herein is a new and selective synthesis of three types of dihydropyrano[3,2-*c*]pyrazoles, namely **7**, **8**, and **20**, with pyrazole-fused heterocyclic skeletons from **1** via the combination of Claisen rearrangements and RCM, along with efforts toward furo[3,2-*c*]pyrazoles (**17**).

## 2. Results

### 2.1. Synthesis of *1*,*5*-Dihydropyrano[*3*,*2*-c]pyrazoles

Our initial efforts in the synthesis of 1,5-dihydropyrano[3,2-*c*]pyrazoles (**7**) are presented in Scheme 1 and Table 1. In our earlier efforts for double bond migration for the conversion of **2a** to **5a** with potassium tert-butoxide (*t*-BuOK) as a base, every trial under microwave (MW) irradiation in a different solvent (tetrahydrofuran (THF), EtOH, MeCN, acetone, 1,2-dimethoxyethane (DME), toluene, THF-toluene) failed to give the desired product **5a** [20,26]. Alternatively, carbonylchlorohydridotris(triphenylphosphine)ruthenium(II) [(RuClH(CO)(PPh_3_)_3_] was applied to the double bond migration for the conversion of **2** to **5**, as shown in Scheme 1 [27]. MW irradiation of the reaction mixture of **2** and 5 mol% of the ruthenium hydride catalyst in toluene gave the desired product **5**, whereas the same reaction at room temperature (rt) did not occur. Starting compounds **2a**–**d** are known compounds [21,22,23], and 1-benzyl-4-hydroxy-5-((1-methoxycarbonyl)-2-propen-1-yl)-1H-pyrazole (**2e**) is the Claisen rearrangement product of **1e**, which was newly prepared from 1-benzyl-4-iodo-1H-pyrazole for this work and already contained a small part of **5e** (see Experimental section).

Then, the C4-hydroxyl groups in 4-hydroxy-5-(1-propenyl)-1H-pyrazoles **5a** and **5b** were treated with aqueous NaOH followed by alkenyl halides in order to prepare the RCM substrates **6a** and **6b**. Conversion of **5c** and **5d**, which have a substituent, to **6c** and **6d** using the same condition took a long time with poor yields. So, alternative transformation of **5c** and **5d** to **6c** and **6d** was carried out using K_2_CO_3_ in acetone under MW irradiation, respectively. The reactions proceeded smoothly and the chemical yields of **6c** and **6d** are presented in Scheme 1a. In a separate experiment, compound **2e**, which already contains a small part of **5e** as noted above, was transformed directly to **6e** through treatment with K_2_CO_3_ and allyl bromide in acetone under MW irradiation in 63% yield, since the yield from **2e** to **5e** was not satisfactory. The yield of the MW-aided transformation of **2e** to **6e** was improved to 85% by applying acetone-water (9:1) as the solvent system (Scheme 1b).

RCM substrates **6** were treated with 5 mol% Grubbs′ second-generation catalyst (Grubbs^2nd^) in CH_2_Cl_2_. The results of the RCM reactions are summarized in Table 1. With substrate **6a**, reaction at rt afforded the desired RCM product **7a** within 30 min (entry 2). A shorter reaction time also led to **7a**, but with an inseparable trace amount of **6a** (entry 1). In contrast, extended reaction times led to reduced product yields (entries 3 and 4). The MW-aided reaction was also examined in an attempt to reduce the reaction time (entries 5–7). In these trials, only **7a** was formed and double bond migration product **8a** could not be detected [23]. Moreover, higher temperatures above 100 °C reduced the reaction yield (entry 7). The optimal reaction conditions in entries 2 and 5 for substrate **6a** were applied to the RCM of **6b** and gave similar results producing **7b** (entries 8 and 9, respectively). The MW reaction of **6b** at a higher temperature of 140 °C led to partial double bond migration to produce **8b** (entry 10). When the substrate had an R′ substituent, different results were obtained, as shown by the following entries. Substrates **6d** and **6e** did not react at rt (entries 13 and 15, respectively). 

The MW-aided reaction (140 °C) of **6c** afforded RCM product **7c** as a minor product (24%) and **9c** (45%) with an exomethylene moiety as the major product (entry 12). The structure of **9c** was determined through the heteronuclear single quantum coherence (HSQC) correlations between a carbon signal at *δ* 107.2 ppm and two proton signals at *δ* 4.78 and 4.96 ppm. Generally, endo-cyclic alkene is considered to be more stable than the corresponding exo-alkene. But in this case, **7c** is thought to be less stable than exo-diene **9c** due to the strain caused by 6-membered endo-diene structure in the thermodynamic condition.

However, the same MW conditions applied to substrate **6d** did not result in **7d**, but dimeric **10d** formed through intermolecular metathesis in 30% yield (entry 14). Mass spectrometry (MS) revealed that compound **10d** had an *m*/*z* of 632 (M^+^), which corresponds to C_42_H_42_N_4_O_2_. The ^1^H nuclear magnetic resonance (NMR) spectrum of **10d** suggested the presence of a =CHCH_3_ moiety through the signals at δ 6.29 (q, *J* = 7.1 Hz) and 1.51 ppm (d, *J* = 7.1 Hz) in a 1:3 integral ratio and the lack of an exomethylene from the starting **6d**. These data suggest that the intermolecular metathesis product **10d** formed by expelling an ethylene molecule [339 (**6d**) × 2 − 28 (CH_2_=CH_2_) = 632 ((M^+^) for **10d**)]. The presence of a bulky R′ substituent may lead to serious repulsion in the transition state for RCM. When the substrate had a methoxycarbonyl group as R′, the results were confusing. The MW reaction of **6e** at 140 °C gave a complex mixture and only **7e** was isolated in 15% yield (entry 18). The MW reactions of **6e** at lower temperatures (80 and 100 °C) gave **10e** in similar yields (29% and 30%, respectively) with **7e** as a minor product (entries 16 and 17). In both of these entries, **11e**, which is a metathesis product of **6e** and the Grubbs catalyst, was also isolated as a minor product. The structure of **11e** was confirmed through detailed NMR analysis and an M^+^ peak at *m*/*z* 388.1785 (C_24_H_24_N_2_O_3_) in the high-resolution MS (HRMS) spectrum. However, our attention was focused on increasing the yields of **7e** and decreasing the yields of **10e** by increasing the reaction temperature (entries 16–18). Then, we hypothesized that **10e** transforms into **7e**; **10e** may be the initial product at lower reaction temperatures. Therefore, the MW reaction of pure **10e** with Grubbs^2nd^ at 140 °C was examined independently in an attempt to observe the formation of **7e** as the major product in the reaction mixture.

### 2.2. Synthesis of *1*,*7*-Dihydropyrano[*3*,*2*-c]pyrazoles

We attempted to expand this methodology to the syntheses of different types of pyrazole-fused heterobicycles, i.e., 1,7-dihydropyrano[3,2-*c*]pyrazoles (**8**) and furo[3,2-*c*] pyrazoles (**17**), as illustrated in Scheme 2. In order to realize this, 4-*O*-vinylation was required. First, the 4-hydroxyl group of **2a** was treated with 1,2-dichloroethane to obtain a pyrazole with a 2-chloroethoxy group at C4, **12_Cl_**. However, dehydrochlorination of **12_Cl_** did not occur under basic conditions. Then, 2-bromoethylation of the 4-hydroxyl group was examined, aimed at improving the leaving ability. Desired 5-allyl-4-(2-bromoethyl)oxy-1*H*-1-tritylpyrazole (**12a**) was smoothly prepared through the MW-aided reaction of **2a**. The examination of the dehydrobromination of **12a** is summarized in Table 2. Whereas treatment of **12a** with *t*-BuOK in toluene resulted in no reaction (entry 1), application of THF-MeOH (4:1) led to the desired dehydrobromination (entries 2–5). The MW reaction at 100 °C for 30 min afforded only double bond migration product (*E*/*Z*)-5-allyl-4-vinyloxy-1*H*-1-tritylpyrazole (**13a**) but in 14% yield (entry 2). Increasing the reaction time to 60 min resulted in an inseparable mixture of **13a** and 5-(1-propenyl)-4-vinyloxy-1*H*-1-tritylpyrazole (**14a**) in 19% combined yield (entry 3). A higher temperature of 130 °C resulted in only **14a** in 30% yield (entry 4). A similar MW reaction at 80 °C produced **13a** in a similar yield (entry 5). In these trials (entries 2–5), the chemical yields of desired **13a** and **14a** were not satisfactory. Close inspection of entries 4 and 5 led us to isolate and elucidate the structures of side product **15** (28% yield), which should have formed via S_N_2 attack by a methoxide on **12a**, and **16** (17% yield) (see footnotes of Table 2). To improve the chemical yields, inhibition of the S_N_2 attack on **12a** by a nucleophile formed from the solvent under basic conditions was required. Hence, *t*-BuOH was applied instead of MeOH as a co-solvent. Although the MW reaction at 80 °C afforded only a trace amount of desired product **13a** (entry 6), the same reaction at 130 °C afforded only **13a** in 87% yield (entry 7). Inspired by the result in entry 4, the MW reaction was attempted at a higher temperature of 180 °C and afforded **14a** selectively in 67% yield (entry 8). Treatment of the *N*-benzyl derivative **12b** with *t*-BuOK at 130 °C resulted in only **14b** (72%) (entry 9). Then, the dehydrobromination of **12b** was examined at a lower temperature (entry 10), but resulted in an inseparable mixture of **12b** and **14b**. 

The RCM of prepared substrates **13a**, **14a**, and **14b** were examined. Treatment of **13a** with Grubbs^2nd^ (5 mol%) at rt gave the desired product **8a** in 95% yield. However, the corresponding reactions of **14a** and **14b** did not afford the desired products **17a** and **17b**, even with MW assistance. Further examinations of **14a** with alternative catalysts, such as the Grubbs^1st^, Hoveyda-Grubbs, and Schrock catalysts, also did not lead to **17a**. Our synthesis of **17** will be continued in a future study.

### 2.3. Synthesis of *2*,*5*-Dihydropyrano[*3*,*2*-c]pyrazoles

The synthesis of 2,5-dihydropyrano[3,2-*c*]pyrazoles (**20**) was examined and the results are summarized in Scheme 3. For this purpose, selective preparation of 3-alkenyl-4-allyloxy-1*H*-pyrazoles **19** is required since 3-allyl-4-hydroxy-1*H*-1-tritylpyrazole is a minor Claisen rearrangement product of **1a**, and the corresponding 3-allyl-1-benzyl-4-hydroxy-1*H*-pyrazole could not be obtained by heating **1b** [21,22]. Hence, an alternative method of preparing **19** via a deprotection-reprotection sequence was examined. 4-Allyloxy-5-(1-propenyl)-1*H*-1-tritylpyrazole (**6a**) was deprotected with aqueous HCl to give **18**, which was then treated with trityl chloride or benzyl bromide under basic conditions. An *E*/*Z* mixture of 4-allyloxy-3-(1-propenyl)-1*H*-1-tritylpyrazole (**19a**) was obtained exclusively owing to the steric repulsion between the propenyl group on the pyrazole ring and an introduced bulky trityl group. However, *N*-benzylation of **18** afforded a mixture of **19b** and **6b** in a ca. 4:1 ratio in 60% combined yield, and separation gave pure **19b** in 25% yield. The obtained substrates **19a** and **19b** were independently treated with 5 mol% Grubbs^2nd^ at rt to afford the desired RCM products **20a** and **20b**, respectively, in good yields.

## 3. Conclusions

We synthesized 1,5-, 1,7-, and 2,5-dihydropyrano[3,2-*c*]pyrazoles (**7**, **8**, and **20**) from 5-allyl-4-hydroxy-1*H*-1-tritylpyrazoles via a combination of the Claisen rearrangement, ruthenium-hydride-catalyzed double bond isomerization, *O*-alkenylation, and RCM. In the synthesis of 1,5-dihydropyrano[3,2-*c*]pyrazoles **7**, the presence of a substituent on the 5-alkenyl group inhibited smooth RCM through steric hindrance. In these cases, MW-aided reactions were effective, but gave various products. Towards the selective synthesis of 1,7-dihydro-1-tritylpyrano[3,2-*c*]pyrazole **8a**, temperature-dependent selective dehydrobromination was effective for preparing the RCM substrate **13b**. For the synthesis of 2,5-dihydropyrano[3,2-*c*]pyrazoles **20**, a deprotection-reprotection sequence was applied to obtain the RCM substrate **19**.

## 4. Materials and Methods

Infrared (IR) spectra were obtained using a Perkin Elmer 1720X FT-IR spectrometer (Perkin Elmer, Wattham, MA, USA). HRMS was performed using a JEOL JMS-700 (2) mass spectrometer (JEOL, Tokyo, Japan). NMR spectra were recorded at 27 °C using Agilent 300, 400-MR-DD2, and 600-DD2 spectrometers in CDCl_3_ using tetramethylsilane (TMS) as the internal standard. Liquid column chromatography was conducted using silica gel BW127ZH (Fuji Silysia Chemical Ltd., Tokyo, Japn). Analytical and preparative thin layer chromatography (TLC) analyses were performed using pre-coated Merck glass plates (silica gel 60 F_254_), and the compounds were visualized by dipping the plates in an ethanol solution of phosphomolybdic acid followed by heating (Merk & Co., Inc., Darmstadt, Germany). MW-assisted reactions were carried out using a Biotage Initiator^®^ (Basel, Switzerland). Anhydrous CH_2_CH_2_ was purchased from Wako Pure Chemical Industries (Osaka, Japan).

### 4.1. Synthesis of (E)-Methyl 4-((1-Benzyl-1H-pyrazol-4-yl)oxy)but-2-enoate (***1e***)

To 1-benzyl-4-formyl-1*H*-pyrazole (200 mg, 1.07 mmol) in CH_2_Cl_2_ (5 mL) was added 70% *meta*-chloroperoxybenzoic acid (397.6 mg, 1.61 mmol) at 0 °C. After it was stirred overnight at room temperature, the mixture was quenched by adding aqueous NaHCO_3_ and then extracted with CH_2_Cl_2_. The organic layer was dried over MgSO_4_, filtered, and evaporated to give a crude residue. The crude material was dissolved in *t*-BuOH-CH_2_Cl_2_ (5 mL/5 mL) at 40 °C, and then potassium *tert*-butoxide (428.6 mg, 3.82 mmol) was added to the solution. After it was stirred overnight at 40 °C, the mixture was quenched with saturated aqueous NH_4_Cl and extracted with CH_2_Cl_2_. The separated organic layer was dried over MgSO_4_, filtered, and evaporated under reduced pressure to afford a crude residue, which was purified using silica gel column chromatography (eluent: EtOAc:hexane = 1:3) to afford (*E*/*Z*)-**1e** (128.1 mg, 44%): oil; IR (film) *v*_max_ 1724 (C=O), 1574 (C=C), 1437 (C=C) cm^−1^; ^1^H NMR (400 MHz, CDCl_3_): δ 3.74 (3H, s, -COO*Me*), 4.52 (2H, dd, *J* = 4.1, 1.9 Hz, -OC*H*_2_CH=CH-), 6.13 (1H, br d, *J* = 15.9 Hz, -COC*H*=CH-), 6.99 (1H, dt, *J* = 15.9, 4.1 Hz, -CH_2_CH=C*H*-), 7.05 (1H, d, *J* = 0.6 Hz, pyrazole-H), 7.18 (2H, br d, *J* = 8.0 Hz, Bn-H), 7.30–7.35 (4H, m, Bn-H, pyrazole-H); ^13^C NMR (100 MHz, CDCl_3_): δ 51.6, 56.6, 70.0, 115.1, 121.4, 127.2, 127.5, 128.0, 128.7, 136.3, 142.5, 145.2, 166.4; high-resolution electron ionization mass spectrometry (HREIMS) *m*/*z* calcd. for C_15_H_16_N_2_O_3_ (M^+^) 272.1161, found 272.1163.

*(*E*)-Methyl 4-((1-trityl-1*H*-pyrazol-4-yl)oxy)but-2-enoate (**1f**) was synthesized in a similar manner as **1e**, but it was not rearranged under the thermal condition described below. **1f**: colorless crystals (CH_2_Cl_2_); mp 155–158 °C; IR (film) *v*_max_ 1725 (C=O), 1572 (C=C), 1492 (C=C), 1442 (C=C) cm^−1^; ^1^H NMR (400 MHz, CDCl_3_): δ 3.76 (3H, s, -COO*Me*), 4.53 (2H, dd, *J* = 4.1, 2.0 Hz, -OC*H*_2_CH=CH-), 5.20 (2H, s, ArC*H*_2_Ph), 6.14 (1H, dt, *J* = 15.9, 1.9 Hz, -COCH=CH-), 7.00 (1H, dt, *J* = 15.9, 4.1 Hz, -CH_2_CH=CH-), 7.05 (1H, s, pyrazole-H), 7.13–7.18 (6H, m, Tr-H), 7.30–7.35 (9H, m, Tr-H), 7.42 (1H, s, pyrazole-H); ^13^C NMR (100 MHz, CDCl_3_): δ 51.7, 70.0, 78.7, 118.4, 121.5, 127.68, 127.71, 127.9, 130.1, 142.5, 143.0, 143.8, 166.4; HREIMS *m*/*z* calcd. for C_27_H_24_N_2_O_3_ (M^+^) 424.1786, found 424.1779.

### 4.2. Synthesis of Methyl 2-(1-Benzyl-4-hydroxy-1H-pyrazol-5-yl)but-3-enoate (***2e***) 

A sealed microwave vial containing a solution of **1e** (128.1 mg, 0.47 mmol) in 1,2-dimethoxyethane (DME) (2 mL) was heated under microwave irradiation at 200 °C for 30 min. After it had cooled, the reaction mixture was quenched with saturated aqueous NH_4_Cl and extracted with CH_2_Cl_2_. The separated organic layer was dried over MgSO_4_, filtered, and evaporated under reduced pressure to afford a crude residue, which was purified using silica gel column chromatography (eluent: EtOAc:hexane = 1:1) to afford **2e** with a small amount of the isomer, **5e** (53.9 mg, 42%). 

**2e** (major) and **5e** (minor) in ca. 2:1 ratio: oil; IR (film) *v*_max_ 1716 (C=O), 1497 (C=C), 1435 (C=C) cm^−1^; ^1^H NMR (400 MHz, CDCl_3_): δ 1.65 (1H, d, *J* = 7.3 Hz, =CHCH_3_ of **5e**), 3.64 (1H, s, -COO*Me* of **5e**), 3.68 (2H, s, -COO*Me* of **2e**), 4.37 (0.7H, br d, *J* = 7.0 Hz, ArC*H*(COOMe)CH=), 4.93 (0.7H, dd, *J* = 17.0, 1.5 Hz, -CH=C*H*H), 5.01 (0.6H, s, ArC*H*_2_Ph), 5.12 (0.7H, dd, *J* = 10.3, 1.5 Hz, -CH=CH*H* of **2e**), 5.19 (0.7H, br d, *J* = 16.1 Hz, ArC*H*HPh of **2e**), 5.25 (0.7H, br d, *J* = 16.1 Hz, ArCH*H*Ph of **2e**), 5.86 (1H, ddd, *J* = 17.0, 10.3, 6.5 Hz, -CH(COOMe)C*H*=CH_2_ of **2e**), 6.83 (0.6H, br s, -OH of **2e**), 7.03–7.05 (2H, m, Ph-H), 7.19–7.31 (3H, m, Ph-H; 0.3H, m, overlapped, =C*H*CH_3_ of **5e**), 7.30 (1H, br s, pyrazole-H); ^13^C NMR (150 MHz, CDCl_3_): δ 15.8 (**5e**), 46.4, 52.3 (**5e**), 53.2, 54.4 (**5e**), 54.6, 118.7, 120.6, 122.4 (**5e**), 122.9 (**5e**), 126.6, 127.2 (**5e**), 127.6 (**5e**), 127.9, 128.2 (**5e**), 128.40 (**5e**), 128.44, 129.1, 130.7, 136.9 (**5e**), 139.8 (**5e**), 140.9, 147.0, 166.5 (**5e**), 173.2; HREIMS *m*/*z* calcd. for C_15_H_16_N_2_O_3_ (M^+^) 272.1161, found 272.1162.

### 4.3. Double Bond Migration of 5-Allyl-4-hydroxy-1H-pyrazoles (Scheme 1)

General procedure: To a toluene solution (10 mL) of 5-allyl-4-hydroxy-1-trityl-1*H*-pyrazole (**2a**) (0.434 g, 1.19 mmol) in a microwave vial (5–20 mL), RuClH(CO)(PPh_3_)_3_ (56.6 mg, 0.059 mmol) was added. The reaction vial was sealed and then heated at 150 °C for 15 min under microwave irradiation. The cooled reaction mixture was evaporated to give a crude residue, which was purified using column chromatography (eluent: hexane:EtOAc = 1:1) to afford 4-hydroxy-5-(1-propenyl)-1-trityl-1*H*-pyrazole (**5a**) (0.323 g, 74% yield) as an *E*/*Z* mixture. 

**Pure starting material gave the desired product as described above, but a small contamination inhibited the isomerization. In that case, a toluene-MeOH (9:1) solvent system was effective for isolating the desired product. 

**5a**: oil; IR (film) *v*_max_ 3268 (-OH), 1597 (C=C), 1494 (C=C), 1446 (C=C) cm^−1^; ^1^H NMR (400 MHz, CDCl_3_): δ 1.43 (3H, dd, *J* = 6.5, 1.2 Hz, =CHC*H*_3_), 5.17 (1H, dd, *J* = 11.4, 1.4 Hz, ArC*H*=CH-), 5.23 (1H, dq, *J* = 11.4, 6.6 Hz, -CH=C*H*CH_3_), 7.09–7.34 (16H, m, Tr-H, pyrazole-H); ^13^C NMR (100 MHz, CDCl_3_): δ 14.9, 78.8, 118.3, 126.2, 127.3, 127.4, 127.6, 127.8, 129.6, 130.0, 130.1, 130.28, 130.34, 142.6; HREIMS *m*/*z* calcd. for C_25_H_22_N_2_O (M^+^) 366.1732, found 366.1731.

(*E*/*Z*)-1-Benzyl-4-hydroxy-5-(1-propenyl)-1*H*-pyrazole (**5b**): *E*/*Z* mixture in ca. 5:1 ratio (X)**;** oil; IR (film) *v*_max_ 3031 (-OH), 1589 (C=C), 1496 (C=C), 1454 (C=C) cm^−1^; ^1^H NMR (600 MHz, CDCl_3_): δ 1.72 (0.5H, dd, *J* = 6.8, 1.5 Hz, -CH=CHC*H*_3_ of (*E*)-isomer), 1.83 (2.5H, dd, *J* = 6.8, 1.8 Hz, -CH=CHC*H*_3_ of (*Z*)-isomer), 5.15 (0.3H, s, -NC*H*_2_Ph of (*Z*)-isomer), 5.24 (1.5H, s, -NC*H*_2_Ph of (*E*)-isomer), 5.93 (0.15H, dq, *J* = 10.1, 6.8 Hz, -CH=C*H*CH_3_ of (*Z*)-isomer), 5.99 (0.15H, dq, *J* = 10.1, 1.5 Hz, ArC*H*=CHCH_3_ of (*Z*)-isomer), 6.15 (0.85H, dq, *J* = 16.1, 1.5 Hz, ArC*H*=CHCH_3_ of (*E*)-isomer), 6.38 (0.85H, dq, *J* = 16.1, 6.8 Hz, -CH=C*H*CH_3_ of (*E*)-isomer), 7.06–7.09 (6H, m, Tr-H), 7.16 (1H, s, pyrazole-H), 7.22–7.31 (9H, m, Tr-H); ^13^C NMR of (*E*)-isomer (150 MHz, CDCl_3_): δ 19.2, 53.7, 116.7, 126.6, 127.6, 127.9, 128.7, 130.3, 137.1, 138.9 (two carbon signals were deduced to have overlapped); (*Z*)-isomer: δ 15.4, 54.0, 115.4, 126.8, 126.9, 127.9, 128.6, 133.4, 137.0, 138.4 (two carbon signals were deduced to have overlapped); HREIMS *m*/*z* calcd. for C_13_H_14_N_2_O (M^+^) 214.1106, found 214.1104.

(*E*/*Z*)-1-Benzyl-4-hydroxy-5-(1-(1-methyl)propen-1-yl)-1*H*-pyrazole (**5c**): *E*/*Z* ratio = ca. 1:1; oil; IR (film) *v*_max_ 3063 (OH), 1563 (C=C), 1497 (C=C), 1456 (C=C) cm^−1^; ^1^H NMR of *E*/*Z* mixture (400 MHz, CDCl_3_): δ 1.41 (1.6H, dd, *J* = 6.9, 1.5 Hz, -CH=CHC*H*_3_), 1.73 (1.4H, dd, *J* = 6.8, 1.2 Hz, -CH=CHC*H*_3_), 1.79 (3H, br s, C_q_CH_3_), 4.30 (0.47H, br s, -OH), 4.43 (0.53H, br s, -OH), 5.08 (0.9H, s, -NC*H*_2_Ph), 5.15 (1.1H, s, -NC*H*_2_Ph), 5.57 (0.47H, qq, *J* = 6.8, 1.6 Hz, -CCH=C*H*CH_3_), 5.78 (0.53H, qq, *J* = 6.9, 1.6 Hz, -CCH_3_=C*H*CH_3_), 7.03 (0.94H, br d, *J* = 6.7 Hz, Ph-H), 7.07 (1.06H, br d, *J* = 6.7 Hz, Ph-H), 7.06–7.09 (6H, m, Ph-H), 7.16–7.36 (3H, m, Ph-H), 7.21 (1H, s, pyrazole-H); ^13^C NMR of *E*/*Z* mixture (100 MHz, CDCl_3_): δ 14.0, 15.0, 16.2, 32.0, 53.9, 54.1, 124.2, 124.6, 126.8, 127.2, 127.4, 127.5, 127.91, 127.94, 128.46, 128.49, 128.6, 129.8, 130.0, 132.6, 137.3, 137.6, 137.8; HREIMS *m*/*z* calcd. for C_14_H_16_N_2_O (M^+^) 228.1263, found 228.1260.

(*E*/*Z*)-1-Benzyl-4-hydroxy-5-(1-(1-phenyl)propenyl)-1*H*-pyrazole (**5d**): isomer ratio = ca. 5:1; oil; IR (film) *v*_max_ 3031 (OH), 1573 (C=C), 1496 (C=C) cm^−1^; ^1^H NMR (100 MHz, CDCl_3_): δ 1.55 (2.5H, d, *J* = 7.0 Hz, =CHC*H*_3_), 1.85 (0.5H, d, *J* = 7.3 Hz, =CHC*H*_3_), 4.61 (1H, br s, *J* = 14.8 Hz, -NC*H*HPh), 4.95 (1H, br s, *J* = 15.3 Hz, -NCH*H*Ph), 5.94 (0.16H, q, *J* = 7.2 Hz, C_q_=C*H*CH_3_), 6.31 (0.84H, br q, *J* = 7.0 Hz, C_q_=C*H*CH_3_), 6.87–6.90 (0.66H, m, Ph-H), 6.93–6.96 (1.34H, m, Ph-H), 7.06–7.34 (4H, m, Ph-H), 7.32 (1H, s, pyrazole-H); ^13^C NMR (100 MHz, CDCl_3_): δ 15.7, 54.3, 126.2, 127.0, 127.38, 127.45, 127.6, 128.3, 128.4, 128.6, 128.9, 129.3, 131.1, 136.9, 139.9 (minor isomer: 15.4, 54.0, 126.4, 127.68, 127.8); HREIMS *m*/*z* calcd. for C_19_H_18_N_2_O (M^+^) 290.1419, found 290.1417.

(*E*/*Z*)-1-Benzyl-4-hydroxy-5-(1-(1-methoxycarbonyl)propenyl)-1*H*-pyrazole (**5e**): *E*/*Z* mixture in ca. 13:1 ratio; oil; IR (film) *v*_max_ 3090 (OH), 1716 (C=O), 1507 (C=C), 1436 (C=C) cm^−1^; ^1^H NMR (400 MHz, CDCl_3_): δ 1.72 (2.6H, d, *J* = 7.3 Hz, -CH=CHC*H*_3_ of major isomer), 1.83 (0.4H, d, *J* = 7.2 Hz, -CH=CHC*H*_3_ of minor isomer), 3.66 (2.6H, s, -OC*H*_3_ of major isomer), 3.71 (0.4H, s, -OC*H*_3_ of minor isomer), 4.73 (1H, s, -OH), 5.06 (1.86H, s, -NC*H*_2_Ph of major isomer), 5.17 (0.14H, s, -NC*H*_2_Ph of minor isomer), 7.04 (2H, br d, *J* = 6.6 Hz, Ph-H), 7.19–7.30 (4H, m, Ph-H, =C*H*CH_3_), 7.33 (1H, s, pyrazole-H); ^13^C NMR (100 MHz, CDCl_3_): δ 15.8, 52.3, 54.5, 122.2, 122.9, 127.3, 127.6, 128.1, 128.4, 136.7, 140.0, 147.5, 166.5; HREIMS *m*/*z* calcd. for C_15_H_16_N_2_O_3_ (M^+^) 272.1161, found 272.1160.

### 4.4. O-Allylation of 1-Protected 5- or 3-Allyl-4-allyloxy-1H-pyrazoles (Scheme 1)

General procedure: To a solution of an *E*/*Z* mixture of 4-hydroxy-5-(1-propenyl)-1*H*-1-tritylpyrazole (**5a**) (0.410 g, 1.12 mmol) in acetone (2 mL), 20% aqueous NaOH (1 mL) and allyl bromide (142 μL, 1.68 mmol) were added. The reaction mixture was stirred for 1 h and then quenched with saturated aqueous NH_4_Cl and extracted with CH_2_Cl_2_. The organic layer was dried over anhydrous MgSO_4_, filtered, and evaporated. The crude residue was purified with column chromatography (eluent: hexane:EtOAc = 3:1) to afford 4-allyloxy-5-(1-propenyl)-1*H*-1-tritylpyrazole (**6a**) (*E*/*Z* mixture in ca. 3:1 ratio, 0.334 g, 73% yield). 

(*E*)-**6a**: mp 152–155 °C; IR (KBr) *v*_max_ 1567 (C=C), 1491 (C=C), 1446 (C=C) cm^−1^; ^1^H NMR (400 MHz, CDCl_3_): δ 1.40 (3H, dd, *J* = 6.7, 1.5 Hz, C*H*_3_CH=), 4.50 (2H, dt, *J* = 5.3, 1.5 Hz, -OC*H*_2_CH=CH_2_), 5.26 (1H, dq, *J* = 15.8, 1.4 Hz, -CH_2_CH=C*H*H), 5.37 (1H, dq, *J* = 17.2, 1.5 Hz, -CH_2_CH=C*H*H), 5.46 (1H, dq, *J* = 15.8, 1.4 Hz, ArC*H*=CHCH_3_), 6.04 (1H, ddt, *J* = 17.2, 10.5, 5.3 Hz, -OCH_2_C*H*=CH_2_), 6.14 (1H, dq, *J* = 15.8, 6.7 Hz, ArCH=C*H*CH_3_), 7.07–7.16 (6H, m, Tr-H), 7.24–7.39 (9H, m, Tr-H), 7.32 (1H, s, pyrazole-H); ^13^C NMR (100 MHz, CDCl_3_): δ 18.9, 72.0, 79.0, 117.5, 120.0, 124.7, 127.3, 127.4, 127.6, 128.5, 130.3, 133.5, 142.9, 143.5; HREIMS *m*/*z* calcd. for C_28_H_26_N_2_O (M^+^) 406.2055, found 406.2047. 

(*Z*)-**6a**: mp 82–86 °C; IR (KBr) *v*_max_ 1567 (C=C), 1491 (C=C), 1446 (C=C) cm^−1^; ^1^H NMR (400 MHz, CDCl_3_): δ 1.42 (3H, d, *J* = 5.1 Hz, C*H*_3_CH=), 4.47 (2H, dt, *J* = 5.5, 1.6 Hz, -OC*H*_2_CH=CH_2_), 5.16–5.20 (2H, m), 5.22 (1H, dq, *J* = 10.6, 1.4 Hz, -CH_2_CH=C*H*H), 5.34 (1H, dq, *J* = 17.2, 1.5 Hz, -CH_2_CH=C*H*H), 6.00 (1H, ddt, *J* = 17.2, 10.6, 5.6 Hz, -OCH_2_C*H*=CH_2_), 7.03–7.17 (6H, m, Tr-H), 7.21–7.32 (9H, m, Tr-H), 7.37 (1H, s, pyrazole-H); ^13^C NMR (100 MHz, CDCl_3_): δ 15.4, 72.0, 78.9, 117.4, 117.9, 124.9, 127.2, 127.3, 127.9, 129.8, 130.1, 133.7, 142.6, 142.9; HREIMS *m*/*z* calcd. for C_28_H_26_N_2_O (M^+^) 406.2046, found 406.2050.

(*E*/*Z*)-4-Allyloxy-1-benzyl-5-(1-propenyl)-1*H*-pyrazole (**6b**) (an inseparable *E*/*Z* mixture in a ca. 8:2 ratio, 0.334 g, 73% yield): oil; IR (film) *v*_max_ 1566 (C=C), 1495 (C=C), 1452 (C=C) cm^−1^; HREIMS *m*/*z* calcd. for C_16_H_18_N_2_O (M^+^) 254.1419, found 254.1421. (*E*)-isomer: ^1^H NMR (600 MHz, CDCl_3_): δ 1.82 (3H, d, *J* = 6.7, 1.8 Hz, C*H*_3_CH=), 4.50 (2H, dt, *J* = 5.4, 1.4 Hz, -OC*H*_2_CH=CH_2_), 5.26 (1H, dq, *J* = 10.6, 1.4 Hz, -CH_2_CH=C*H*H), 5.28 (2H, s, NC*H*_2_Ph), 5.39 (1H, ddd, *J* = 17.3, 3.2, 1.8 Hz, -CH_2_CH=CH*H*), 6.05 (1H, ddt, *J* = 17.3, 10.6, 5.4 Hz, -OCH_2_C*H*=CH_2_), 6.16 (1H, br d, *J* = 15.8 Hz, ArC*H*=CHCH_3_), 6.46 (1H, dq, *J* = 15.8, 6.7 Hz, -CH=C*H*CH_3_), 7.07 (2H, br d, *J* = 7.6 Hz, Ph-H), 7.24 (1H, br t, *J* = 7.6 Hz, Ph-H), 7.26 (1H, s, pyrazole-H), 7.30 (2H, br t, *J* = 7.6 Hz, Ph-H); ^13^C NMR (150 MHz, CDCl_3_): δ 19.3, 53.9, 72.2, 116.6, 117.5, 125.8, 126.5, 126.8, 127.5, 128.6, 128.7, 129.6, 133.5, 137.2; (*Z*)-isomer: ^1^H NMR (600 MHz, CDCl_3_): δ 1.72 (3H, dd, *J* = 6.8, 1.8 Hz, C*H*_3_CH=), 4.46 (2H, dt, *J* = 5.6, 1.5 Hz, -OC*H*_2_CH=CH_2_), 5.18 (2H, s, NC*H*_2_Ph), 5.24 (1H, dq, *J* = 10.5, 1.5 Hz, -CH_2_CH=C*H*H), 5.36 (1H, dq, *J* = 17.1, 1.5 Hz, -CH_2_CH=CH*H*), 5.91 (1H, dq, *J* = 11.2, 6.8 Hz, -CH=C*H*CH_3_), 6.01 (1H, ddt, *J* = 17.1, 10.5, 5.6 Hz, -OCH_2_C*H*=CH_2_), 6.01 (1H, br d, *J* = 11.2 Hz, ArC*H*=CHCH_3_, overlapped), 7.07 (2H, br d, *J* = 7.6 Hz, Ph-H), 7.24 (1H, br t, *J* = 7.6 Hz, Ph-H), 7.28 (1H, s, pyrazole-H), 7.30 (2H, br t, *J* = 7.6 Hz, Ph-H); ^13^C NMR (150 MHz, CDCl_3_): δ 15.7, 53.9, 72.4, 115.4, 117.4, 126.59, 126.64, 127.5, 133.2, 137.1, 142.7 (three signals should be overlapped with signals of the (*E*)-isomer).

(*E*/*Z*)-4-Allyloxy-1-benzyl-5-(1-(1-methyl)propenyl)-1*H*-pyrazole (**6c**): isomer ratio = ca. 1:1; oil; IR (film) *v*_max_ 1562 (C=C), 1496 (C=C), 1455 (C=C) cm^−1^; ^1^H NMR (400 MHz, CDCl_3_): δ 1.43 (1.6H, dd, *J* = 6.7, 1.4 Hz, C*H*_3_CH=), 1.72 (1.4H, dd, *J* = 6.9, 1.2 Hz, C*H*_3_CH=), 4.414 (1.06H, d, *J* = 5.5 Hz, -OC*H*_2_CH=), 4.417 (0.94H, d, *J* = 5.5 Hz, -OC*H*_2_CH=), 5.08 (0.94H, s, NC*H*_2_Ph), 5.19 (1.06H, s, NC*H*_2_Ph), 5.19–5.36 (2H, m, =CH_2_), 5.54 (0.44H, qq, *J* = 6.9, 1.6 Hz, -C(CH_3_)*H*=CH_3_), 5.75 (0.56H, qq, *J* = 6.8, 1.6 Hz, -C(CH_3_)*H*=CH_3_), 5.92–6.04 (1H, m, -CH_2_C*H*=CH_2_), 7.02 (0.94H, br d, *J* = 7.3 Hz, Ph-H), 7.07 (1.06H, br d, *J* = 7.2 Hz, Ph-H), 7.18–7.29 (3H, m, Ph-H), 7.29 (1H, s, pyrazole-H); ^13^C NMR (100 MHz, CDCl_3_): δ 13.9, 15.2, 16.0, 23.1, 53.7, 54.0, 72.8, 117.45, 117.54, 125.0, 126.77, 126.86, 127.21, 127.3, 127.5, 128.46, 128.49, 129.0, 129.2, 129.5, 133.2, 133.7, 133.8, 137.3, 137.8, 141.6, 141.7; HREIMS *m*/*z* calcd. for C_17_H_20_N_2_O (M^+^) 268.1576, found 268.1575.

(*E*/*Z*)-4-Allyloxy-1-benzyl-5-(1-(1-phenyl)propenyl)-1*H*-pyrazole (**6d**): isomer ratio = ca. 7:1; oil; IR (film) *v*_max_ 1556 (C=C), 1500 (C=C) cm^−1^; ^1^H NMR (400 MHz, CDCl_3_): δ 1.55 (2.7H, d, *J* = 6.9 Hz, C*H*_3_CH=), 1.85 (0.3H, d, *J* = 7.3 Hz, C*H*_3_CH=), 4.38 (0.25H, br d, *J* = 5.5 Hz, -OC*H*_2_CH=), 4.44 (1.75H, br d, *J* = 3.9 Hz, -OC*H*_2_CH=CH_2_), 4.64 (1H, br d, *J* = 14.9 Hz, NC*H*HPh), 4.94 (1H, br d, *J* = 14.8 Hz, NCH*H*Ph), 5.19 (1H, dd, *J* = 10.5, 1.3 Hz, -CH_2_CH=C*H*H), 5.29 (1H, dq, *J* = 17.2, 1.6 Hz, -CH_2_CH=CH*H*), 5.89–6.99 (1H, m, -OCH_2_C*H*=CH_2_ overlaps with 0.12H, m, =C*H*CH_3_), 6.31 (0.88H, q, *J* = 7.0 Hz, =C*H*CH_3_), 6.89–6.90 (2H, m, Ph-H), 7.08–7.49 (8H, m, Ph-H), 7.39 (1H, s, pyrazole-H); ^13^C NMR (100 MHz, CDCl_3_): δ 15.8, 54.3, 72.7, 117.5, 126.2, 127.0, 127.1, 127.4, 128.27, 128.33, 128.5, 129.2, 129.3, 131.2, 133.7, 137.0, 139.7, 143.3; HREIMS *m*/*z* calcd. for C_22_H_22_N_2_O (M^+^) 330.1732, found 330.1729.

Synthesis of (*E*/*Z*)-4-allyloxy-1-benzyl-5-(1-(1-methoxycarbonyl)propenyl)-1*H*-pyrazole (**6e**) from **2e**: To an acetone solution (4.5 mL) of **2e** with a small amount of **5e** (121.8 mg, 0.45 mmol) in a microwave vial were added K_2_CO_3_ (61.8 mg, 0.45 mmol) in water (0.5 mL) and allyl bromide (0.04 mL, 0.45 mmol). After the reaction vial was sealed, the mixture was heated under microwave irradiation at 60 °C for 1 h. After it had cooled, the reaction was quenched by adding aqueous NH_4_Cl. Then, the reaction mixture was extracted with EtOAc three times. The organic layer was washed with brine, dried over MgSO_4_, filtered, and then evaporated to give a crude residue, which was purified using column chromatography (eluent: hexane:EtOAc = 2:1) to give pure **6e** (117.1 mg, 84%). 

**6e** (isomer ratio = ca. 13:1): oil; IR (film) *v*_max_ 1717 (C=O), 1500 (C=C) cm^−1^; ^1^H NMR: δ 1.54 (2.6H, d, *J* = 7.2 Hz, C*H*_3_CH= of major isomer), 2.12 (0.4H, d, *J* = 7.2 Hz, C*H*_3_CH= of minor isomer), 3.52 (0.4H, s, -OC*H*_3_ of minor isomer), 3.60 (2.6H, s, -OC*H*_3_ of major isomer), 4.92 (0.93H, br d, *J* = 15.3 Hz, NCH*H*Ph of major isomer), 5.01 (0.14H, s, NC*H*_2_Ph of minor isomer), 5.03 (0.93H, br d, *J* = 15.3 Hz, NC*H*HPh of major isomer), 5.11 (0.93H, dq, *J* = 10.6, 1.4 Hz, -CH_2_CH=C*H*H of major isomer), 5.14 (0.07H, dq, *J* = 10.6, 1.5 Hz, -CH_2_CH=C*H*H of minor isomer), 5.22 (0.93H, dq, *J* = 17.5, 1.6 Hz, -CH_2_CH=CH*H* of major isomer), 5.23 (1H, dq, *J* = 17.4, 1.6 Hz, -CH_2_CH=CH*H* of minor isomer), 5.87–5.93 (1H, m, -OCH_2_C*H*=CH_2_), 6.46 (0.07H, q, *J* = 7.3 Hz, -C_q_=C*H*CH_3_ of minor isomer), 7.06 (2H, br d, *J* = 6.6 Hz, Ph-H), 7.17–7.30 (3H, m, Ph-H), 7.20 (0.07H, q, *J* = 7.2 Hz, -C_q_=C*H*CH_3_ of major isomer), 7.32 (1H, s, pyrazole-H); ^13^C NMR (100 MHz, CDCl_3_): δ 15.7, 52.0, 54.6, 72.5, 117.6, 122.0, 123.2, 126.4, 127.4, 127.6, 128.4, 133.4, 136.7, 143.2, 147.9, 165.9 (minor isomer: 16.2, 51.5, 54.1, 72.8, 121.5, 123.1, 126.6, 127.5, 136.9, 147.7); HREIMS *m*/*z* calcd. for C_18_H_20_N_2_O_3_ (M^+^) 312.1474, found 312.1467.

### 4.5. Ring-Closing Metathesis of 6 to 1H-1,5-Dihydropyrano[*3,2*-c]pyrazoles ***7*** (Table 1)

General procedure (Table 1, entry 3): To a solution of **6a** (21.8 mg, 0.054 mmol) in CH_2_Cl_2_ (2 mL) was added Grubbs^2nd^ (1.7 mg, 2.7 mmol) at rt. The reaction mixture was stirred at rt for 1 h, and then the solvent was removed under reduced pressure, affording a crude residue, which was purified using silica gel column chromatography (eluent: EtOAc:hexane = 1:3) to afford **7a** (16.2 mg, 83%).

*General procedure for MW-aided reaction (Table 1, entry 5): To a solution of **6a** (16.4 mg, 0.04 mmol) in CH_2_Cl_2_ (2 mL) was added Grubbs^2nd^ (2.3 mg, 2.0 mmol) in a microwave vial. The reaction mixture was heated under microwave irradiation at 80 °C for 3 min. After the reaction mixture had cooled, the solvent was removed under reduced pressure, affording a crude residue, which was purified using silica gel column chromatography (eluent: EtOAc:hexane = 1:4) to afford **7a** (12.8 mg, 87%).

1,5-Dihydro-1-tritylpyrano[3,2-*c*]pyrazole (**7a**): oil; IR (film) *v*_max_ 1677 (C=C), 1581 (C=C), 1493 (C=C), 1447 (C=C) cm^−1^; ^1^H NMR (400 MHz, CDCl_3_): δ 4.64 (2H, dd, *J* = 3.8, 1.8 Hz, -OC*H*_2_CH=CH-), 5.15 (1H, dt, *J* = 10.2, 3.7 Hz, -OCH_2_C*H*=CH-), 5.28 (1H, dtd, *J* = 10.2, 1.8, 0.8 Hz, -OCH_2_CH=C*H*-), 7.08–7.17 (6H, m, Tr-H), 7.18 (1H, d, *J* = 0.8 Hz, pyrazole-H), 7.23–7.32 (9H, m, Tr-H); ^13^C NMR (100 MHz, CDCl_3_): δ 66.9, 78.0, 117.7, 118.6, 124.3, 127.55, 127.57, 130.1, 141.4, 142.7; HREIMS *m*/*z* calcd. for C_25_H_20_N_2_O (M^+^) 364.1575, found 364.1585.

1-Benzyl-1,5-dihydropyrano[3,2-*c*]pyrazole (**7b**): oil; IR (film) *v*_max_ 1566 (C=C), 1495 (C=C), 1452 (C=C) cm^−1^; ^1^H NMR (400 MHz, CDCl_3_): δ 4.75 (2H, dd, *J* = 3.9, 1.8 Hz, -OC*H*_2_CH=), 5.21 (2H, s, ArC*H*_2_Ph), 5.53 (1H, dt, *J* = 10.0, 3.9 Hz, -OCH_2_C*H*=CH-), 6.34 (1H, br d, *J* = 10.0 Hz, -OCH_2_CH=C*H*-), 7.10 (1H, d, *J* = 0.8 Hz, pyrazole-H), 7.10–7.14 (2H, d, *J* = 6.6 Hz, Ph-H), 7.26–7.32 (3H, m, Ph-H); ^13^C NMR (100 MHz, CDCl_3_): δ 54.0, 67.2, 115.5, 119.7, 124.5, 127.1, 127.9, 128.8, 136.6, 140.9; HREIMS *m*/*z* calcd. for C_13_H_12_N_2_O (M^+^) 212.0950, found 212.0949.

1-Benzyl-1,5-dihydro-7-methylpyrano[3,2-*c*]pyrazole (**7c**): oil; IR (film) *v*_max_ 1732 (C=O), 1541 (C=C) cm^−1^; ^1^H NMR (400 MHz, CDCl_3_): δ 1.96 (3H, br s, C_q_C*H*_3_), 4.64 (2H, dq, *J* = 3.3, 1.6 Hz, -OC*H*_2_C*H*=), 5.23–5.26 (1H, m, -OCH_2_C*H*=), 5.39 (2H, s, NC*H*_2_Ph), 7.01 (2H, br d, *J* = 7.0 Hz, Ph-H), 7.17 (1H, s, pyrazole-H), 7.24–7.32 (3H, m, Ph-H); ^13^C NMR (100 MHz, CDCl_3_): δ 18.1, 55.3, 67.6, 116.5, 124.7, 126.0, 127.6, 127.3, 128.7, 137.7, 141.4; HREIMS *m*/*z* calcd. for C_15_H_14_N_2_O_3_ (M^+^) 270.1004, found 270.1003.

1-Benzyl-1,5-dihydro-7-methoxycarbonylpyrano[3,2-*c*]pyrazole (**7e**): oil; IR (film) *v*_max_ 1732 (C=O), 1541 (C=C) cm^−1^; ^1^H NMR (400 MHz, CDCl_3_): δ 3.76 (3H, s, -COOC*H*_3_), 4.72 (2H, d, *J* = 4.5 Hz, -OC*H*_2_CH=), 5.57 (2H, s, ArC*H*_2_Ph), 6.45 (1H, t, *J* = 4.5 Hz, -OCH_2_C*H*=C_q_), 6.33 (1H, br d, *J* = 10.0 Hz, -OCH_2_CH=C*H*-), 7.04 (2H, br d, *J* = 6.5 Hz, Ph-H), 7.23 (1H, *s*, pyrazole-H), 7.23–7.31 (3H, m, Ph-H); ^13^C NMR (100 MHz, CDCl_3_): δ 52.3, 56.4, 66.9, 124.0, 124.7, 127.0, 127.4, 128.4, 128.7, 137.5, 142.4, 163.8; HREIMS *m*/*z* calcd. for C_1__5_H_14_N_2_O_3_ (M^+^) 270.1004, found 270.1003.

1-Benzyl-1,7-dihydropyrano[3,2-*c*]pyrazole (**8b**): oil; IR (film) *v*_max_ 1607 (C=C), 1586 (C=C), 1557 (C=C) cm^−1^; ^1^H NMR (400 MHz, CDCl_3_): δ 3.23 (2H, dd, *J* = 3.3, 2.0 Hz, ArC*H*_2_CH=), 4.77 (1H, dt, *J* = 6.3, 3.4 Hz, -CH_2_C*H*=CH-), 5.17 (2H, s, *Ar*C*H*_2_Ph), 6.42 (1H, dt, *J* = 6.2, 2.0 Hz, =CH=C*H*O-), 7.06–7.20 (2H, m, Ph-H), 7.22–7.33 (4H, m, Ph-H, pyrazole-H); ^13^C NMR (100 MHz, CDCl_3_): δ 19.5, 53.8, 97.1, 125.1, 126.4, 127.0, 127.9, 128.8, 129.0, 136.6, 141.3; HREIMS *m*/*z* calcd. for C_13_H_12_N_2_O (M^+^) 212.0950, found 212.0947.

1-Benzyl-1,7-dihydro-7-methylenepyrano[3,2-*c*]pyrazole (**9c**): oil; IR (film) *v*_max_ 1644 (C=C), 1556 (C=C), 1401 (C=C) cm^−1^; ^1^H NMR (400 MHz, CDCl_3_): δ 2.54 (2H, br t, *J* = 5.6 Hz, -OCH_2_C*H*_2_C_q_), 4.17 (2H, t, *J* = 5.7 Hz, -OC*H*_2_CH_2_-), 4.78 (1H, br s, C_q_C*H*H), 4.96 (1H, br s, C_q_CH*H*), 5.43 (2H, s, NC*H*_2_Ph), 7.02 (2H, d, *J* = 7.0 Hz, Ph-H), 7.22–7.32 (4H, m, Ph-H, pyrazole-H); ^13^C NMR (100 MHz, CDCl_3_): δ 32.2, 55.4, 68.3, 107.2, 124.0, 125.3, 126.2, 127.5, 128.7, 129.7, 136.9, 142.6; HREIMS *m*/*z* calcd. for C_14_H_14_N_2_O (M^+^) 226.1106, found 226.1102.

1,4-Bis((1-benzyl-5-(1-phenylprop-1-en-1-yl)-1*H*-pyrazol-4-yl)oxy)but-2-ene (**10d**): oil; IR (film) *v*_max_ 1569 (C=C), 1496 (C=C) cm^−1^; ^1^H NMR (400 MHz, CDCl_3_): δ 1.51 (6H, d, *J* = 7.1 Hz, =CHC*H*_3_), 4.40 (4H, br s, -OC*H*_2_CH=), 4.62 (2H, br d, *J* = 14.8 Hz, *Ar*C*H*HPh), 4.92 (2H, br d, *J* = 14.4 Hz, *Ar*CH*H*Ph), 5.82–5.84 (2H, m, -OCH_2_C*H*=), 6.29 (2H, q, *J* = 7.1 Hz, =C*H*CH_3_), 6.92–6.95 (4H, m, Ph-H), 7.06–7.25 (6H, m, Ph-H), 7.35 (2H, s, pyrazole-H); ^13^C NMR (100 MHz, CDCl_3_): δ 15.8, 54.3, 71.6, 126.1, 127.1, 127.35, 127.42, 128.3, 128.7, 129.1, 129.2, 131.2, 137.0, 140.0, 143.2 (three carbon signals overlapped); HREIMS *m*/*z* calcd. for C_42_H_40_N_4_O_2_ (M^+^) 632.3151, found 632.3145.

1,4-Bis((1-benzyl-5-(1-(methoxycarbonyl)prop-1-en-1-yl)-1*H*-pyrazol-4-yl)oxy)but-2-ene (**10e**): oil; IR (film) *v*_max_ 1722 (C=O), 1712 (C=O), 1642 (C=C), 1573 (C=C) cm^−1^; ^1^H NMR (400 MHz, CDCl_3_): δ 1.54 (5.4H, d, *J* = 7.0 Hz, =CHC*H*_3_ of major isomer), 2.14 (0.6H, d, *J* = 7.2 Hz, =CHC*H*_3_ of minor isomer), 3.61 (0.6H, s, -OC*H*_3_ of minor isomer), 3.62 (5.4H, s, =CHC*H*_3_ of major isomer), 4.42 (3.6H, br s, -OC*H*_2_CH= of major isomer), 4.48 (0.4H, br s, -OC*H*_2_CH= of minor isomer), 5.08 (1.8H, br d, *J* = 13.3 Hz, ArC*H*HPh of major isomer), 5.11 (0.4H, s, ArC*H*_2_Ph of minor isomer), 5.12 (1.8H, br d, *J* = 13.3 Hz, ArCH*H*Ph of major isomer), 5.77 (3.6H, br t, *J* = 3.7 Hz, -OCH_2_C*H*= of minor isomer), 5.88 (0.4H, br t, *J* = 3.7 Hz, -OCH_2_C*H*= of major isomer), 6.28 (0.2H, q, *J* = 7.5 Hz, =C*H*CH_3_ of minor isomer), 7.08 (4H, d, *J* = 6.8 Hz, Ph-H), 7.20–7.32 (7.8H, m, Ph-H, =C*H*CH_3_ of major isomer), 7.33 (2H, s, pyrazole-H); ^13^C NMR (100 MHz, CDCl_3_): δ 15.7, 52.1, 54.6, 71.5, 122.1, 123.2, 126.4, 127.4, 127.6, 128.4, 128.5, 136.7, 147.9, 165.9; HREIMS *m*/*z* calcd. for C_34_H_36_N_4_O_6_ (M^+^) 596.2635, found 596.2634.

Methyl 2-(1-benzyl-4-(cinnamyloxy)-1*H*-pyrazol-5-yl)but-2-enoate (**11e**): oil; IR (film) *v*_max_ 1716 (C=O), 1644 (C=C), 1574 (C=C) cm^−1^; ^1^H NMR (600 MHz, CDCl_3_): δ 1.58 (3H, d, *J* = 7.3 Hz, =CHC*H*_3_), 3.58 (3H, s, -COOC*H*_3_), 4.58 (2H, d, *J* = 6.2 Hz, -OC*H*_2_CH=), 5.02 (1H, br d, *J* = 15.2 Hz, ArC*H*HPh), 5.12 (1H, br d, *J* = 15.2 Hz, ArCH*H*Ph), 6.30 (1H, dt, *J* = 15.9, 6.2 Hz, -OCH_2_C*H*=CH-), 6.62 (1H, d, *J* = 15.9 Hz, -CH=C*H*Ph), 7.09 (2H, d, *J* = 7.3 Hz, Ph-H), 7.20–7.38 (8H, m, Ph-H), 7.30 (1H, q, *J* = 7.3 Hz, -C_q_=C*H*CH_3_), 7.40 (1H, s, pyrazole-H); ^13^C NMR (150 MHz, CDCl_3_): δ 15.8, 52.0, 54.7, 72.7, 122.2, 123.6, 124.7, 126.6, 126.9, 127.4, 127.6, 127.9, 128.4, 128.6, 133.1, 136.4, 136.7, 143.2, 147.9, 165.9; HREIMS *m*/*z* calcd. for C_24_H_24_N_2_O_3_ (M^+^) 388.1787, found 388.1785.

### 4.6. Synthesis of 5-Allyl-4-(2-haloethoxy)-1H-pyrazoles (***12***) (Scheme 2)

General procedure: To a solution of **2a** (50.8 mg, 0.14 mmol) in acetone (2 mL) in a microwave vial were added 1,2-dibromoethane (0.05 mL, 0.56 mmol), 20% aqueous NaOH (0.11 mL, 0.56 mmol), and a catalytic amount of tetrabutylammonium bromide. The sealed reaction vial was MW irradiated at 140 °C for 30 min. After it had cooled, the reaction mixture was quenched with saturated aqueous NH_4_Cl and extracted with CH_2_Cl_2_. The separated organic layer was dried over MgSO_4_, filtered, and evaporated under reduced pressure to afford a crude residue. The residue was purified using silica gel column chromatography (eluent: EtOAc:hexane = 1:3) to afford **12a** (42.9 mg, 65%) as an oil. 

5-Allyl-4-(2-bromoethoxy)-1*H*-1-tritylpyrazole (**12a**): pale yellow crystals (CH_2_Cl_2_); mp 135–140 °C; IR (film) *v*_max_ 1581 (C=C), 1491 (C=C), 1446 (C=C) cm^−1^; ^1^H NMR (500 MHz, CDCl_3_): δ 2.85 (2H, dt, *J* = 6.5, 1.2 Hz, ArC*H*_2_CH=CH_2_), 3.56 (2H, t, *J* = 6.2 Hz, -OCH_2_C*H*_2_CBr), 4.20 (2H, t, *J* = 6.2 Hz, -OC*H*_2_CH_2_Br), 4.63 (1H, dq, *J* = 17.0, 1.6 Hz, -CH=CH*H*), 4.66 (1H, dq, *J* = 10.0, 1.4 Hz, -CH=CH*H*), 4.97 (1H, ddt, *J* = 17.0, 10.0, 6.5 Hz, -CH_2_C*H*=CH_2_), 7.10–7.13 (6H, m, Tr-H), 7.25–7.30 (9H, m, Tr-H), 7.33 (1H, s, pyrazole-H); ^13^C NMR (125 MHz, CDCl_3_): δ 29.4, 31.2, 71.6, 78.7, 115.9, 125.6, 127.4, 127.6, 129.9, 130.1, 132.4, 142.8, 143.6; HREIMS *m*/*z* calcd. for C_27_H_25_BrN_2_O (M^+^) 472.1151, found 472.1149.

5-Allyl-1-benzyl-4-(2-bromoethoxy)-1*H*-pyrazole (**12b**): oil; IR (film) *v*_max_ 1583 (C=C), 1496 (C=C) cm^−1^; ^1^H NMR (400 MHz, CDCl_3_): δ 3.29 (2H, dd, *J* = 4.7, 1.7 Hz, ArC*H*_2_CH=), 3.56 (2H, br t, *J* = 6.2 Hz, -OCH_2_C*H*_2_Br), 4.20 (2H, br t, *J* = 6.2 Hz, -OC*H*_2_CH_2_Br), 5.00 (1H, dd, *J* = 7.0, 1.4 Hz, -CH=CH*H*), 5.07 (1H, dd, *J* = 10.2, 1.4 Hz, -CH=CH*H*), 5.73–5.83 (1H, m, -CH_2_C*H*=CH_2_), 7.06 (2H, br d, *J* = 8.1 Hz, Bn-H), 7.25–7.33 (4H, m, Ph-H, pyrazole-H); ^13^C NMR (100 MHz, CDCl_3_): δ 27.1, 29.5, 53.9, 72.4, 116.5, 126.7, 127.2, 127.7, 127.8, 128.7, 133.6, 136.9, 141.6; HREIMS *m*/*z* calcd. for C_15_H_17_BrN_2_O (M^+^) 320.0524, found 320.0520.

5-Allyl-4-(2-chloroethoxy)-1*H*-1-tritylpyrazole (**12_Cl_**): white powder (CH_2_Cl_2_); mp 120–125 °C; IR (KBr) *v*_max_ 1580 (C=C), 1493 (C=C), 1446 (C=C) cm^−1^; ^1^H NMR (500 MHz, CDCl_3_): δ 2.85 (2H, br d, *J* = 7.6 Hz, ArC*H*_2_CH=), 3.72 (2H, t, *J* = 5.7 Hz, -OCH_2_C*H*_2_Cl), 4.14 (2H, t, *J* = 5.7 Hz, -OC*H*_2_CH_2_CCl), 4.63 (1H, dq, *J* = 17.0, 1.6 Hz, -CH=CH*H*), 4.66 (1H, dq, *J* = 10.0, 1.4 Hz, -CH=CH*H*), 4.97 (1H, ddt, *J* = 17.0, 10.0, 6.7 Hz, -CH_2_C*H*=CH_2_), 7.10–7.14 (6H, m, Tr-H), 7.24–7.31 (9H, m, Tr-H), 7.34 (1H, s, pyrazole-H); ^13^C NMR (125 MHz, CDCl_3_): δ 31.2, 42.1, 71.7, 78.6, 115.8, 125.5, 127.3, 127.6, 129.9, 130.0, 132.4, 142.8, 143.7; HREIMS *m*/*z* calcd. for C_27_H_25_ClN_2_O (M^+^) 428.1655, found 428.1654. *MW conditions: 160 °C, 30 min.

### 4.7. Reaction of ***12*** with Potassium Tert-Butoxide (Table 2, Scheme 2)

General procedure (Table 2, entry 7): To a solution of **12a** (28.8 mg, 0.05 mmol) in anhydrous THF:*t*-BuOH (2 mL:0.5 mL) in a microwave vial was added potassium *tert*-butoxide (28.8 mg, 0.26 mmol). The sealed reaction vial was MW irradiated at 130 °C for 1 h. After it had cooled, the reaction mixture was quenched with saturated aqueous NH_4_Cl and extracted with CH_2_Cl_2_. The separated organic layer was dried over MgSO_4_, filtered, and evaporated under reduced pressure to afford a crude residue. The residue was purified using silica gel column chromatography (eluent: EtOAc:hexane = 1:3) to afford **13a** (20.8 mg, 87%).

5-Allyl-1-trityl-1*H*-4-vinyloxypyrazole (**13a**): white powder (CH_2_Cl_2_); mp 75–80 °C; IR (KBr) *v*_max_ 1639 (C=C), 1624 (C=C), 1566 (C=C) cm^−1^; ^1^H NMR (600 MHz, CDCl_3_): δ 2.81 (2H, ddd, *J* = 6.8, 1.5, 1.2 Hz, ArC*H*_2_CH=CH_2_), 4.23 (1H, dd, *J* = 5.4, 1.8 Hz, -OCH=C*H*H), 4.50 (1H, dd, *J* = 13.8, 2.1 Hz, -OCH_2_=CH*H*), 4.62 (1H, dq, *J* = 16.7, 1.5 Hz, -CH_2_CH=C*H*H), 4.68 (1H, dq, *J* = 10.9, 1.5 Hz, -CH_2_CH=CH*H*), 4.99 (1H, ddt, *J* = 16.5, 10.9, 2.1 Hz, -CH_2_C*H*=CH_2_), 6.53 (1H, dd, *J* = 13.8, 6.5 Hz, -OC*H*=CH_2_), 7.12–7.14 (6H, m, Tr-H), 7.25–7.31 (9H, m, Tr-H), 7.40 (1H, s, pyrazole-H); ^13^C NMR (150 MHz, CDCl_3_): δ 31.1, 77.8, 91.3, 116.1, 127.4, 127.6, 128.3, 130.0, 131.7, 131.9, 140.4, 142.3, 150.7; HREIMS *m*/*z* calcd. for C_27_H_24_N_2_O (M^+^) 392.1888, found 392.1880.

5-(1-Propenyl)-1-trityl-1*H*-4-vinyloxypyrazole (**14a**): white powder (CH_2_Cl_2_); mp 133–135 °C; IR (KBr) *v*_max_ 1639 (C=C), 1560 (C=C), 1492 (C=C) cm^−1^; ^1^H NMR (600 MHz, CDCl_3_): δ 1.39 (3H, dd, *J* = 6.8, 1.8 Hz, -CH=CHC*H*_3_), 4.29 (1H, dd, *J* = 6.2, 2.0 Hz, -OCH=C*H*H), 4.59 (1H, dd, *J* = 13.8, 2.0 Hz, -OCH=CH*H*), 5.39 (1H, br dq, *J* = 15.8, 0.8 Hz, -C*H*=CHCH_3_), 5.98 (1H, dq, *J* = 15.8, 6.8 Hz, -CH=C*H*CH_3_), 6.56 (1H, dd, *J* = 13.8, 6.2 Hz, -OC*H*=CH_2_), 7.11–7.15 (6H, m, Tr-H), 7.26–7.32 (9H, m, Tr-H), 7.38 (1H, br s, pyrazole-H); ^13^C NMR (150 MHz, CDCl_3_): δ 18.8, 79.8, 92.3, 119.1, 127.38, 127.44, 128.0, 129.1, 130.3, 131.2, 139.5, 142.7, 150.4; HREIMS *m*/*z* calcd. for C_27_H_24_N_2_O (M^+^) 392.1889, found 392.1887.

(*E*/*Z*)-1-Benzyl-5-(1-propenyl)-1*H*-4-vinyloxypyrazole (**14b**): *E*/*Z* ratio = ca. 5:1; oil; IR (film) *v*_max_ 1642 (C=C), 1562 (C=C), 1493 (C=C) cm^−1^; ^1^H NMR (400 MHz, CDCl_3_): δ 1.68 (0.5H, d, *J* = 6.3 Hz, =CHC*H*_3_ of (*Z*)-isomer), 1.82 (2.5H, dd, *J* = 6.6, 1.6 Hz, =CHC*H*_3_ of (*E*)-isomer), 4.22 (0.17H, dd, *J* = 6.3, 2.0 Hz, -CH=C*H*H of (*Z*)-isomer), 4.29 (0.83H, dd, *J* = 6.3, 2.0 Hz, -CH=C*H*H of (*E*)-isomer), 4.57 (0.17H, dd, *J* = 13.7, 2.0 Hz, -CH=CH*H* of (*Z*)-isomer), 4.59 (0.83H, dd, *J* = 13.7, 2.0 Hz, -CH=CH*H* of (*E*)-isomer), 5.93 (0.17H, dq, *J* = 11.0, 6.5 Hz, -CH=C*H*CH_3_ of (*Z*)-isomer), 5.98 (0.17H, br d, *J* = 11.0 Hz, ArC*H*=CHCH_3_ of (*Z*)-isomer), 6.11 (0.83H, br dq, *J* = 16.0, 1.6 Hz, ArC*H*=CHCH_3_ of (*E*)-isomer), 6.34 (0.83H, dq, *J* = 15.8, 6.8 Hz, -CH=C*H*CH_3_ of (*E*)-isomer), 6.49 (0.17H, dd, *J* = 13.7, 6.3 Hz, -OC*H*=CH_2_ of (*Z*)-isomer), 6.55 (0.83H, dd, *J* = 13.7, 6.3 Hz, -OC*H*=CH_2_ of (*E*)-isomer), 7.07 (2H, br d, *J* = 7.0 Hz, Ph-H), 7.23–7.37 (3H, m, Ph-H), 7.33 (1H, br s, pyrazole-H); ^13^C NMR (150 MHz, CDCl_3_): δ 16.0 (minor), 19.3, 53.4 (minor), 54.0, 91.9 (minor), 92.3, 114.7 (minor), 115.9, 126.5, 126.9, 127.7, 128.7, 128.8, 129.0 (minor), 131.5, 134.2 (minor), 136.9, 138.5 (minor), 150.4 (minor), 150.5; HREIMS *m*/*z* calcd. for C_15_H_16_N_2_O (M^+^) 240.1263, found 240.1256.

(*E*/*Z*)-4-(2-Methoxy)ethoxy-3-(1-propenyl)-2*H*-2-tritylpyrazole (**15**): oil; IR (film) *v*_max_ 1492 (C=C), 1446 (C=C) cm^−1^; ^1^H NMR (400 MHz, CDCl_3_): δ 1.39 (3H, d, *J* = 6.7 Hz, =CHC*H*_3_), 3.65 (0.5H, br t, *J* = 4.1 Hz, -OC*H*_2_CH_2_Br), 3.7 (1.5H, br t, *J* = 4.1 Hz, -CH_2_C*H*_2_Br), 4.07 (0.5H, br t, *J* = 3.9 Hz, -CH_2_C*H*_2_Br), 3.70 (1.5H, br t, *J* = 4.1 Hz, -OC*H*_2_CH_2_Br), 5.44 (1H, br d, *J* = 15.8 Hz, (*E*)-ArC*H*=CH-), 6.09–6.18 (1H, m, -CH=C*H*CH_3_), 7.11–7.20 (6H, m, Tr-H), 7.24–7.29 (9H, m, Tr-H), 7.32 (1H, s, pyrazole-H); ^13^C NMR (150 MHz, CDCl_3_): δ 18.9, 59.2, 70.6, 71.3, 79.0, 119.9, 124.8, 127.3, 127.4, 127.6, 130.1, 130.4, 142.9, 143.7; HREIMS *m*/*z* calcd. for C_28_H_28_N_2_O_2_ (M^+^) 424.2151, found 424.2157.

4-(2-Methoxy)ethoxy-3-(2-propenyl)-2*H*-2-tritylpyrazole (**16**): oil; IR (film) *v*_max_ 1580 (C=C), 1447 (C=C) cm^−1^; ^1^H NMR (600 MHz, CDCl_3_): δ 2.84 (2H, br d, *J* = 6.5 Hz, ArC*H*_2_CH=), 3.41 (3H, s, -OC*H*_3_), 3.66 (1H, br t, *J* = 5.0 Hz, -OC*H*_2_CH_2_O-), 4.05 (2H, br t, *J* = 5.0 Hz, -OCH_2_C*H*_2_O-), 4.60 (1H, dq, *J* = 17.0, 1.7 Hz, -CH=C*H*H), 4.64 (1H, dq, *J* = 10.5, 1.5 Hz, -CH=CH*H*), 7.10–7.13 (6H, m, Tr-H), 7.23–7.33 (9H, m, Tr-H), 7.34 (1H, s, pyrazole-H); ^13^C NMR (150 MHz, CDCl_3_): δ 31.2, 59.2, 71.2, 71.4, 78.5, 115.6, 125.5, 127.3, 127.6, 127.9, 130.1, 132.6, 143.0, 144.4; HREIMS *m*/*z* calcd. for C_28_H_28_N_2_O_2_ (M^+^) 424.2151, found 424.2157.

### 4.8. RCM of ***13a*** and ***14a*** and ***14b***

The RCM reactions of **13a** and **14a** and **14b** in Scheme 2 were carried out as described above.

1,7-Dihydro-1-tritylpyrano[3,2-*c*]pyrazole (**8a**): oil; IR (film) *v*_max_ 1583 (C=C), 1493 (C=C), 1446 (C=C) cm^−1^; ^1^H NMR (600 MHz, CDCl_3_): δ 2.27 (2H, dd, *J* = 3.5, 2.0 Hz, ArC*H*_2_CH=CH-), 4.49 (1H, dt, *J* = 6.5, 3.5 Hz, -OCH=C*H*CH_2_-), 6.33 (1H, dt, *J* = 6.4, 2.0 Hz, -OC*H*=CHCH_2_-), 7.12–7.15 (6H, m, Tr-H), 7.26–7.32 (9H, m, Tr-H), 7.32 (1H, s, pyrazole-H); ^13^C NMR (150 MHz, CDCl_3_): δ 22.5, 29.7, 78.6, 98.2, 124.3, 127.6, 127.6, 127.9, 130.4, 140.3, 142.6; HREIMS *m*/*z* calcd. for C_25_H_20_N_2_O (M^+^) 364.1575, found 364.1576.

### 4.9. Acid-Catalyzed Hydrolysis of ***6a*** (Scheme 3)

To a solution of **6a** (121.5 mg, 0.30 mmol) in acetone (10 mL) was added 1 N aqueous HCl (0.6 mL). The reaction mixture was warmed under reflux for 90 min with stirring. After the reaction mixture had cooled, it was treated with saturated aqueous NaHCO_3_ and extracted with CH_2_Cl_2_. The separated organic layer was dried over MgSO_4_, filtered, and condensed under reduced pressure to give a crude residue, which was purified using silica gel column chromatography (eluent: EtOAc:hexane = 1:2) to afford (*Z*)-**18** (9.1 mg, 20%) and (*E*)-**18** (14.2 mg, 31%).

(*E*)-4-Allyloxy-5-(1-propenyl)-1*H*-pyrazole ((*E*)-**18**): oil; IR (film) *v*_max_ 1568 (C=C), 1516 (C=C) cm^−1^; ^1^H NMR (600 MHz, CDCl_3_): δ 1.89 (3H, br d, *J* = 6.0 Hz, C*H*_3_CH=), 4.60 (2H, dt, *J* = 5.5, 1.5 Hz, -OC*H*_2_CH=CH_2_), 4.46 (1H, dq, *J* = 10.0, 1.5 Hz, -CH=C*H*H), 5.40 (1H, dq, *J* = 17.3, 1.5 Hz, -CH=CH*H*), 6.04 (1H, ddt, *J* = 17.3, 10.5, 5.5 Hz, -OCH_2_C*H*=CH_2_), 6.34 (1H, d, *J* = 16.7 Hz, ArC*H*=CH-), 6.35–6.41 (1H, m, -CH=C*H*CH_3_), 7.22 (1H, s, pyrazole-H); ^13^C NMR (150 MHz, CDCl_3_): δ 18.9, 72.7, 117.7, 118.7, 127.6, 133.4, 142.0; HREIMS *m*/*z* calcd. for C_9_H_12_N_2_O (M^+^) 164.0950, found 164.0950. 

(*Z*)-4-Allyloxy-5-(1-propenyl)-1*H*-pyrazole ((*Z*)-**18**): oil; IR (film) *v*_max_ 1570 (C=C), 1524 (C=C), 1450 (C=C) cm^−1^; ^1^H NMR (600 MHz, CDCl_3_): δ 1.98 (3H, dd, *J* = 7.0, 1.8 Hz, C*H*_3_CH=), 3.49 (3H, s, -OC*H*_3_), 4.45 (2H, dt, *J* = 5.2, 1.5 Hz, -OC*H*_2_CH=CH_2_), 5.27 (1H, dq, *J* = 10.6, 1.5 Hz, -CH=C*H*H), 5.38 (1H, dq, *J* = 17.0, 1.5 Hz, -CH=CH*H*), 5.82 (1H, dq, *J* = 11.4, 7.0 Hz, -CH=C*H*CH_3_), 6.03 (1H, ddt, *J* = 17.3, 10.6, 5.3 Hz, -OCH_2_C*H*=CH_2_), 6.27 (1H, dq, *J* = 11.5, 1.5 Hz, ArC*H*=CHCH_3_), 7.27 (1H, s, pyrazole-H); ^13^C NMR (150 MHz, CDCl_3_): δ 69.1, 69.3, 117.7, 118.7, 127.6, 133.4, 142.0; HREIMS *m*/*z* calcd. for C_9_H_12_N_2_O (M^+^) 164.0950, found 164.0949. 

### 4.10. Reprotection of ***18*** (Scheme 3)

General procedure: To a stereo mixture of (*E*/*Z*)-**18** (15.9 mg, 0.10 mmol) in CH_2_Cl_2_ (10 mL) were added TrCl (43.0 mg, 0.15 mmol) and Et_3_N (0.022 mL, 0.15 mmol) at 0 °C. The reaction mixture was stirred at rt overnight, and then quenched with saturated aqueous NH_4_Cl and extracted with CH_2_Cl_2_. The organic layer was dried over MgSO_4_, filtered, and condensed under reduced pressure to give a crude residue, which was purified using silica gel column chromatography (eluent: EtOAc:hexane = 1:4) to afford **19a** (28.9 mg, 68%) as an oil.

(*E*/*Z*)-4-Allyloxy-3-(1-propenyl)-1*H*-1-tritylpyrazole (**19a**): oil; IR (film) *v*_max_ 1560 (C=C), 1491 (C=C), 1445 (C=C) cm^−1^; ^1^H NMR of (*E*)-isomer (600 MHz, CDCl_3_): δ 1.90 (3H, dd, *J* = 7.1, 1.8 Hz, C*H*_3_CH=CH-), 4.27 (2H, dt, *J* = 5.6, 1.5 Hz, -OC*H*_2_CH=CH_2_), 5.20 (1H, ddd, *J* = 10.6, 3.2, 1.5 Hz, -CH_2_CH=CH*H*), 5.28 (1H, ddd, *J* = 17.0, 3.2, 1.8 Hz, -CH_2_CH=C*H*H), 5.75 (1H, dq, *J* = 11.5, 7.1 Hz, -CH=C*H*CH_3_), 5.95 (1H, ddt, *J* = 17.3, 10.7, 5.6 Hz, -OCH_2_C*H*=CH_2_), 6.29 (1H, dq, *J* = 11.5, 1.5 Hz, ArC*H*=CHCH_3_), 6.84 (1H, s, pyrazole-H), 7.14–7.18 (6H, m, Tr-H), 7.26–7.30 (9H, m, Tr-H); ^13^C NMR (150 MHz, CDCl_3_): δ (14.2), 15.6, (60.4), 72.9, 78.6, (117.4), 117.6, 117.7, 127.4, 127.5, (127.6), 127.9, 130.4, 133.3, (138.6), (142.0), 143.4, signals in parentheses correspond to some of those of the (*Z*)-isomer; HREIMS *m*/*z* calcd. for C_28_H_26_N_2_O (M^+^) 406.2045, found 406.2040.

(*E*)-4-Allyloxy-1-benzyl-3-(1-propenyl)-1*H*-pyrazole (**19b**): oil; IR (film) *v*_max_ 1566 (C=C), 1496 (C=C), 1445 (C=C) cm^−1^; ^1^H NMR (400 MHz, CDCl_3_): δ 1.87 (3H, dd, *J* = 6.3, 1.2 Hz, C*H*_3_CH=CH-), 4.35 (2H, dt, *J* = 5.4, 1.5 Hz, -OC*H*_2_CH=CH_2_), 5.16 (2H, s, ArC*H*_2_Ph), 5.24 (1H, dq, *J* = 10.5, 1.5 Hz, -CH_2_CH=CH*H*), 5.36 (1H, dq, *J* = 17.2, 1.6 Hz, -CH_2_CH=C*H*H), 6.00 (1H, ddt, *J* = 17.2, 10.5, 5.4 Hz, -OCH_2_C*H*=CH_2_), 6.40 (1H, br d, *J* = 16.3 Hz, ArC*H*=CHCH_3_), 6.53 (1H, dq, *J* = 16.3, 6.3 Hz, ArCH=C*H*CH_3_), 6.92 (1H, s, pyrazole-H), 7.18 (2H, br d, *J* = 8.0 Hz, Ph-H), 7.26–7.35 (3H, m, Ph-H); ^13^C NMR (100 MHz, CDCl_3_): δ 18.9, 56.5, 72.6, 114.7, 117.6, 121.4, 127.4, 127.9, 128.7, 133.2, 136.8, 138.4, 143.1; HREIMS *m*/*z* calcd. for C_16_H_18_N_2_O (M^+^) 254.1419, found 254.1416.

### 4.11. RCM of ***19***

The RCM reactions of **19** were carried out in a similar manner to that described above to afford **20**.

2,5-Dihydro-2-tritylpyrano[3,2-*c*]pyrazole (**20a**): oil; IR (film) *v*_max_ 1492 (C=C), 1447 (C=C) cm^−1^; ^1^H NMR (400 MHz, CDCl_3_): δ 4.76 (2H, dd, *J* = 2.9, 1.9 Hz, -OC*H*_2_CH=CH-), 5.72 (1H, dt, *J* = 10.2, 3.5 Hz, -OCH_2_C*H*=CH-), 6.62 (1H, br d, *J* = 10.0 Hz, -OCH_2_CH=C*H*Ar), 6.80 (1H, s, pyrazole-H), 7.18–7.20 (6H, m, Tr-H), 7.28–7.31 (9H, m, Tr-H); ^13^C NMR (150 MHz, CDCl_3_): δ 67.2, 78.0, 116.5, 120.1, 122.7, 126.5, 127.6, 127.7, 137.5, 139.1, 143.3; HREIMS *m*/*z* calcd. for C_25_H_20_N_2_O (M^+^) 364.1575, found 364.1584.

2-Benzyl-2,5-dihydropyrano[3,2-*c*]pyrazole (**20b**): oil; IR (film) *v*_max_ 1660 (C=C), 1576 (C=C) cm^−1^; ^1^H NMR (400 MHz, CDCl_3_): δ 4.77 (2H, dd, *J* = 3.5, 1.9 Hz, -OC*H*_2_CH=CH-), 5.73 (1H, dt, *J* = 10.0, 3.5 Hz, -OCH_2_C*H*=CH-), 6.63 (1H, dt, *J* = 10.0, 1.9 Hz, -OCH_2_CH=C*H*Ar), 6.84 (1H, s, pyrazole-H), 7.18–7.20 (2H, br d, *J* = 6.5 Hz, Ph-H), 7.27–7.36 (3H, m, Ph-H); ^13^C NMR (100 MHz, CDCl_3_): δ 56.4, 67.2, 113.3, 119.5, 122.3, 127.5, 128.0, 128.8, 136.6, 137.2, 140.5; HREIMS *m*/*z* calcd. for C_13_H_12_N_2_O (M^+^) 212.0949, found 212.0950.

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
