# Peer review of "Synthesis of Dihydropyrano[3,2-c]pyrazoles via Double Bond Migration and Ring-Closing Metathesis"

_molecules, 2019, doi:10.3390/molecules24020296_

Round 1
Reviewer 1 Report
This manuscript well described synthesis of fused pyrazole ring by Claisen rearrangement and ring closing metathesis. The complexity and diversity of polycyclic heterocyclic compound library is highly important in medicinal chemistry. The current method can provide general synthetic methods to achieve synthesis of dihydropyrano[3,2-c] pyrazole. Additionally all experimental details are described to follow easily. Therefore this reviewer recommend for publication in Molecules journal with some revision.
Page 1 line 14 O-allylation O italic
Page 1 line 30 reference 10 was not included in Figure 1
Page 1 line 30 O-allylation O italic
Page 2 Fig 1 number and alphabet for heterocyclic ring should be minimized. Some of them were overlapped with bond.
Page 2 line 42 O-allylation O italic
Page 2 line 56 KOtBu t-BuOK
Page 3 line 68 C4 C4
Page 3 line 69 6a and b 6a and 6b
Page 3 line 82 starting 6a should be 6a
Page 4 Scheme 1 1f: need to be aligned. Reference need to be moved to at the end of text
This scheme is too complicated to understand. Detailed information of substituent would be put once in 1a-1f
Page 4 footnote a, b, c, d need to be superscript. Other text would be 9 points
Page 4 line 103 δ and J should be italic
Page 4 line 112 an M+ peak an should be changed or removed?
Page 5 line 122 4-O-vinylation 4-O-vinylation
Page 5 line 123 C4, 12Cl need change 12a or 12c
Page 5 line 123 12Cl need change 12a or 12c
Page 5 Scheme 2
12Cl need to be changed to another number for example 12a or 12c
KOtBu t-BuOK
Page 5 line 128 and 143 all KOtBu should be changed to t-BuOK
Page 6 table 2 footnote need to be changed as described above
Page 6 line 168 N-benzylation N-benzylation
Page 6 Scheme 3 RX TO TrCl or BnBr, remove bottom (RX: TrCl, BnBr), move Et3N, CH2Cl2 to bottom
Page 6 line 177 O-alkenylation O-alkenylation
In experimental section all δ and J should be italic
Page 7 line 199 the crude formate (it is confusing with formic acid) the crude material?
Page 9 line 288 O-allylation O-allylation
Page 10 line 366 1.7 mg, 0.0027 mmol) 1.7 mg, 2.7 µmol)
Page 10 line 370 2.3 mg, 0.002 µmol)
Page 12 line 450 C27H2579BrN2O 79 need to be removed
Page 12 line 456 C15H1779BrN2O 79 need to be removed
Page 12 line 464 C27H2535ClN2O 35 need to be removed
Page 13 line 517 518 14a and b 14a and 14b
Page 13 line 532 18E (E)-18
Page 14 line 538 18Z (Z)-18
Hyperlink in references need to be removed.
Author Response
Please find the attached files of the revised manuscript “Synthesis of Dihydropyrano[3,2-c]pyrazoles via Double Bond Migration and Ring-Closing Metathesis”. We thank the Editor and the reviewers for many useful advices. We respond and revised manuscript as follows.
Responce to Reviewer 1
Page 1 line 14 O-allylation O italic: done
Page 1 line 30 reference 10 was not included in Figure 1: reference 10 was moved to the preceeding sentence.
Page 1 line 30 O-allylation O italic: line 37 done
Page 2 Fig 1 number and alphabet for heterocyclic ring should be minimized. Some of them were overlapped with bond. : revised as ordered.
Page 2 line 42 O-allylation O italic: done
Page 2 line 56 KOtBu t-BuOK: done
Page 3 line 68 C4 C4: done
Page 3 line 69 6a and b 6a and 6b: done
Page 3 line 82 starting 6a should be 6a : done
Page 4 Scheme 1 1f: need to be aligned. Reference need to be moved to at the end of text;done
This scheme is too complicated to understand. Detailed information of substituent would be put once in 1a-1f; we modified text and Scheme 1.
Page 4 footnote a, b, c, d need to be superscript. Other text would be 9 points; They are superscripted.
Page 4 line 103 δ and J should be italic; they are italic.
Page 4 line 112 an M+ peak an should be changed or removed?; modified as adding M+
Page 5 line 122 4-O-vinylation 4-O-vinylation; revised
Page 5 line 123 12Cl need change 12a or 12c; no change has been made since they are different compounds.
Page 5 Scheme 2 12Cl need to be changed to another number for example 12a or 12c; same as above, KOtBu t-BuOK: revised.
Page 5 line 128 and 143 all KOtBu should be changed to t-BuOK; revised
Page 6 table 2 footnote need to be changed as described above; same as mentioned above
Page 6 line 168 N-benzylation N-benzylation; revised.
Page 6 Scheme 3 RX TO TrCl or BnBr, remove bottom (RX: TrCl, BnBr), move Et3N, CH2Cl2 to bottom; modified as orderd.
Page 6 line 177 O-alkenylation O-alkenylation; revised.
In experimental section all δ and J should be italic; modified.
Page 7 line 199 the crude formate (it is confusing with formic acid) the crude material? ; revised as “ crude material” as ordered.
Page 9 line 288 O-allylation O-allylation; revised.
Page 10 line 366 1.7 mg, 0.0027 mmol) 1.7 mg, 2.7 µmol); revised.
Page 10 line 370 2.3 mg, 0.002 µmol) revised.
Page 12 line 450 C27H2579BrN2O 79 need to be removed: removed
Page 12 line 456 C15H1779BrN2O 79 need to be removed; removed.
Page 12 line 464 C27H2535ClN2O 35 need to be removed; removed.
Page 13 line 517 518 14a and b 14a and 14b; modified.
Page 13 line 532 18E (E)-18 ; revised.
Page 14 line 538 18Z (Z)-18 revised.
Hyperlink in references need to be removed; removed.
Reviewer 2 Report
In this manuscript, synthesis of dihydropyranopyrazoles using Claisen rearrangement, double bond migration, and ring closing metathesis (RCM). Although this method may be similar to that for dihydrooxepinopyrazoles (ref. 23), the present method could provide a novel approach to access dihydropyranopyrazoles. The authors should consider the following points, especially the third comment.
1) RCM reactions of 6a amd 6b at room temperature for 30 min giving 7a (92%, Table 1, entry 2) and 7b (96%, entry 8), respectively are so effective that microwave irradiation conditions are not necessary to be examine (entries 5-7, 9, 10).
2) Normally, an endo-cyclic olefin is more stable than the corresponding exo-cyclic one. However, endo-cyclic olefin 7c underwent isomerization to exo-cyclic olefin 9c under RCM conditions. The authors should give some explanation.
3) The difficulty of RCM of 6d or 6e having bulky R’ group (Ph or CO2Me; Table 1, entries 13-18) may be due to steric interaction between R’ group and N-protective group R (Bn) in the cyclization steps. The authors should try RCM of 3-alkenyl-4-allyloxy-1H-pyrazole derivatives, analogues of 19a and 19b (see Scheme 3).
4) R should be R’ in the formula for compounds 10d and 10e (Table 1).
5) “J” for coupling constants should be typed in italic form (see, page 4).
6) All references exhibit the titles of the articles except ref. 25.
Author Response
Please find the attached files of the revised manuscript “Synthesis of Dihydropyrano[3,2-c]pyrazoles via Double Bond Migration and Ring-Closing Metathesis”. We thank the Editor and the reviewers for many useful advices. We respond and revised manuscript as follows.
Comments and Suggestions for Authors; response are seen in highlighted in blue.
1) RCM reactions of 6a amd 6b at room temperature for 30 min giving 7a (92%, Table 1, entry 2) and 7b (96%, entry 8), respectively are so effective that microwave irradiation conditions are not necessary to be examine (entries 5-7, 9, 10);
Thank you for comment. In our previous study, there were difference between the MW reaction and normal condition. In this trial there was not observed for 6a but difference was observed on 6b at higher temperature. We made some modification in text.
2) Normally, an endo-cyclic olefin is more stable than the corresponding exo-cyclic one. However, endo-cyclic olefin 7c underwent isomerization to exo-cyclic olefin 9c under RCM conditions. The authors should give some explanation.;
We add explanation in text.
3) The difficulty of RCM of 6d or 6e having bulky R’ group (Ph or CO2Me; Table 1, entries 13-18) may be due to steric interaction between R’ group and N-protective group R (Bn) in the cyclization steps. The authors should try RCM of 3-alkenyl-4-allyloxy-1H-pyrazole derivatives, analogues of 19a and 19b (see Scheme 3).;
We are sorry that we cannot respond the reviewer’s requirement. When we try RCM of 3-alkenyl-4-allyloxy-1H-pyrazole derivatives having a substutuent, R is restricted in benzyl group at this time. For deprotection of benzyl group, heterogeneous hydrogenation condition will be applied. Then double bonds in substrates should be reduced at the same time. The products cannot be applied to RCM.
4) R should be R’ in the formula for compounds 10d and 10e (Table 1);
Modified.
5) “J” for coupling constants should be typed in italic form (see, page 4);
Revised to italic.
6) All references exhibit the titles of the articles except ref. 25.;
We added title of reference 25.
Reviewer 3 Report
This manuscript by Usami and coworkers reported their attempts in developing a unified strategy to synthesize a family of of pyrazole fused ring systems via a three-step reaction sequence (Claisen, Alkylation, RCM).
The results are significant and should be of interest to the readers. However, I feel the presentation needs some improvement.
Specifically, the way Scheme 1 and Table 1 are presented is somewhat confusing. In scheme 1, many substrates require different reaction conditions, and the authors have to describe them individually in length in the text. If this is the case, perhaps it's better to list them separately in the Schemes as well.
In Table 2, it's necessary to re-list what a,b,c,d... are.
In the text, the authors sometimes go into detailed deliberation on structure determination. I got the (maybe wrong) impression that in some cases, the results are still inconclusive. I think it is best to be very certain about the structures, and leave the deliberation in the SI, so the main text does not need to be so nitty-gritty.
Finally, the authors noted many failed examples and did not give enough insights into the reasons. The authors also sometimes make a change in the reaction conditions without going into the reasoning. In scientific writing, it's better to give some justification for the observations. Otherwise, it becomes essentially a copy of a lab report instead of a scientific article. For example, some schemes for the proposed routes via which byproducts are formed can be very helpful.
Overall, I think the manuscript needs some revision to make the presentation more clear.
Author Response
Please find the attached files of the revised manuscript “Synthesis of Dihydropyrano[3,2-c]pyrazoles via Double Bond Migration and Ring-Closing Metathesis”. We thank the Editor and the reviewers for many useful advices. We respond and revised manuscript as follows.
response
1. Specifically, the way Scheme 1 and Table 1 are presented is somewhat confusing. In scheme 1, many substrates require different reaction conditions, and the authors have to describe them individually in length in the text. If this is the case, perhaps it's better to list them separately in the Schemes as well.;
We modified Scheme 1 with text but Table 1 is unmodified.
2. In Table 2, it's necessary to re-list what a,b,c,d... are;
They were re-listed as the reviewer’s suggestion.
3. In the text, the authors sometimes go into detailed deliberation on structure determination. I got the (maybe wrong) impression that in some cases, the results are still inconclusive. I think it is best to be very certain about the structures, and leave the deliberation in the SI, so the main text does not need to be so nitty-gritty.;
We are sorry to make no change on structural determination of some compounds. We think it’s minimum explanation on compounds 9c and 11e. Description of the structural determination of dimeric 10d and 10e is important since formation of the undesired dimer seems interesting for the authors.
4. Finally, the authors noted many failed examples and did not give enough insights into the reasons. The authors also sometimes make a change in the reaction conditions without going into the reasoning. In scientific writing, it's better to give some justification for the observations. Otherwise, it becomes essentially a copy of a lab report instead of a scientific article. For example, some schemes for the proposed routes via which byproducts are formed can be very helpful.;
We deleted 1f and 2f from Scheme 1 and text, whereas it was shortly mentioned in Experimental. We add some explanation for changing the reaction condition for Scheme 1. And one more requirement of the proposed route toward byproduct 10d, 10e and 11e was cut along our willing to inhibit taking significant room to be described. We thank the reviewer for many useful suggestions.
Round 2
Reviewer 2 Report
The manuscript has been improved enough to be published in the journal, Molecule.